# Evaluation of Muscle Function by Means of a Muscle-Specific and a Global Index

**DOI:** 10.3390/s21217186

**Published:** 2021-10-29

**Authors:** Samanta Rosati, Marco Ghislieri, Gregorio Dotti, Daniele Fortunato, Valentina Agostini, Marco Knaflitz, Gabriella Balestra

**Affiliations:** Department of Electronics and Telecommunications and PoliToBIOMed Lab, Politecnico di Torino, 10129 Torino, Italy; samanta.rosati@polito.it (S.R.); gregorio.dotti@polito.it (G.D.); daniele.fortunato@polito.it (D.F.); valentina.agostini@polito.it (V.A.); marco.knaflitz@polito.it (M.K.); gabriella.balestra@polito.it (G.B.)

**Keywords:** gait analysis, EMG, muscle activation patterns, movement analysis

## Abstract

Gait analysis applications in clinics are still uncommon, for three main reasons: (1) the considerable time needed to prepare the subject for the examination; (2) the lack of user-independent tools; (3) the large variability of muscle activation patterns observed in healthy and pathological subjects. Numerical indices quantifying the muscle coordination of a subject could enable clinicians to identify patterns that deviate from those of a reference population and to follow the progress of the subject after surgery or completing a rehabilitation program. In this work, we present two user-independent indices. First, a muscle-specific index (MFI) that quantifies the similarity of the activation pattern of a muscle of a specific subject with that of a reference population. Second, a global index (GFI) that provides a score of the overall activation of a muscle set. These two indices were tested on two groups of healthy and pathological children with encouraging results. Hence, the two indices will allow clinicians to assess the muscle activation, identifying muscles showing an abnormal activation pattern, and associate a functional score to every single muscle as well as to the entire muscle set. These opportunities could contribute to facilitating the diffusion of surface EMG analysis in clinics.

## 1. Introduction

The assessment of the muscle activation during human locomotion is necessary to perform a comprehensive gait analysis. In previous studies, instrumented gait analysis proved to be a powerful tool to quantitatively assess muscle activation during locomotion [1,2,3]. In the last decades, the activation of muscles during gait was studied through surface Electromyography (sEMG), which allows for the determination of the timing and extent of muscle activation [4,5,6] without relevant patient discomfort. A typical dynamic sEMG evaluation session consists of three subsequent phases. Phase I: Patient preparation, preliminary tests on the correct positioning of the probes, and patient instruction. Biomedical engineers, physiotherapists, and kinesiologists/human motion scientists usually carry out this phase [7]. Phase II: Signal acquisition, processing, and quality control. Biomedical engineers and kinesiologists/human motion scientists usually carry out this phase [7]. Phase III: Analysis and interpretation of signals and data obtained as output of the previous phase. A multidisciplinary team comprising kinesiologists/human motion scientists, clinical neurophysiologists, physiatrists, biomedical engineers, and physical therapists evaluate signals and data obtained in the previous phase and jointly prepare the clinical report [7].

Phase I: It depends on the ability of operators in: (i) correctly positioning the sEMG probes; (ii) performing a preliminary check of the signal quality; (iii) correctly instructing the patient on how to perform the movement to be studied. SEMG probe positioning may be standardized following existing protocols (i.e., see “Results of the Seniam European project” http://www.seniam.org/ (accessed on 18 October 2021)). Moreover, when looking at muscle activation intervals, a probe displacement as large as 20 mm along the muscle causes a timing error smaller than 1% of the gait cycle. Hence, probe positioning is not a major cause of variability of the results. Preliminary tests on the patient or poor patient instruction may be controlled by using very simple protocols and trained operators. Hence, this phase is not considered as a major cause of variability of the test results.

Phase II: Signal processing methods adopted are the major cause of poor repeatability of results. It has been shown in the past that pre-processing—usually denoising and bandpass filtering—as well as the choice of parameters to obtain the linear envelope of the signal and, in some cases, the activation intervals, are important causes of poor repeatability of results among different gait analysis laboratories. In fact, these are “user-dependent” choices, which can be replaced by automated algorithms that do not require any interaction by operators and that are generally referred to as “user-independent”. Several user-independent algorithms have been published in the past in specialized journals and thoroughly characterized and validated [8,9,10,11,12,13,14,15,16]. These algorithms are aimed at standardizing signal processing methods and signal quality control. The aim is to warrant that results obtained in different laboratories are comparable. The two indices we present in this paper are intended to be used in Phase II to quantify the adherence of the activation pattern of a muscle belonging to a specific subject to the activation prototype of that muscle obtained on a specific reference population as well as in Phase III to facilitate the interpretation of results.

Phase III: Analysis and interpretation of signals and data obtained through Phase II is generally carried out by a team of different professionals. This phase is highly subjective and results strongly depend on the team qualification, which may differ in different laboratories. Nonetheless, to facilitate the exchange of the results of gait analysis sessions among different laboratories, a common standard for test reports should be developed and gain a large consensus. At this time, to our best knowledge, a general consensus has not been reached on any of the report prototypes proposed. The two indices presented in this paper may help the team of professionals carrying out the interpretation of the test results and hence in finding consensus on the coordination of muscle activations.

The high variability of sEMG signals collected during gait, even in healthy subjects, makes it difficult to compare the muscle activity of different subjects and to find similarities or differences that could be of clinical interest [17]. Statistical Gait Analysis (SGA) has been proved to lessen this limitation, through the acquisition and processing of a large number of gait cycles [18,19,20,21]. This methodology allows for an automatic and user-independent analysis of sEMG, goniometric, and foot-switch signals collected during walking sessions lasting several minutes, and hence containing up to some hundreds of strides. In the literature, there is evidence of a great variability of the muscle activation patterns, both intra- and inter-subjects [8,22]. In previous works, the CIMAP (Clustering for Identification of Muscle Activation Patterns) algorithm was proposed to cope with intra-subject variability [9,10]. This algorithm allows for grouping strides showing similar sEMG activation patterns and, as a spin-off of this procedure, to obtain the subject’s Principal Activations (PAs), as the intersection of the cluster prototypes [9,10]. PAs have been defined, from a biomechanical point of view, as those muscle activations that are strictly necessary for accomplishing a specific motor task: they describe the essential contributions of a specific muscle to kinematics, and they are reasonably repeatable among normal subjects [9,10]. This concept is complementary to that of Secondary Activations (SAs), which are activations that have auxiliary functions and are not repeatable during a single walk also within the same subject. The concept of Principal and Secondary Activations applied to the analysis of sEMG signals may significantly simplify the understanding of muscle contribution to the biomechanics of movement, and it has also been applied to the study of muscle synergies [23,24].

The terms “*muscle function*”, “*muscle activation*”, and “*EMG signal*” are crucial for the understanding of basic muscle physiology. The term “*muscle function*” refers to the force production of an active muscle that, in turn, causes its biomechanical action and its contribution to the movement. The term “*muscle activation*” means the physiological active state of a muscle: muscle fibers are activated by the release of acetylcholine underneath the end-plates which, in turn, release the neurotransmitter when they are reached by the depolarization spikes traveling along the second motor neuron that innervates the motor unit the muscle fibers belong to. When a muscle fiber is activated, depolarization arises underneath the end-plate and travels along the muscle fiber, thus causing its contraction. The “*EMG signal*”, that may be detected invasively or by means of surface electrodes (sEMG), is due to the summation of the action potentials of the active muscle fibers that are within the detection volume of the probe. Then, the (s)EMG signal (a physiological variable) is a sign of muscle activation (a physiological state of the muscle) that, in turn, is responsible for the biomechanical action of the muscle (muscle function). There is a large consensus on the fact that sEMG provides information on the neuromuscular function that is not provided by other assessment techniques [7].

In recent years, the objective assessment, based on gait data, of locomotion dysfunctions has become a research field of great interest. In literature, several works showed that it is possible to take advantage of gait parameters to improve the diagnostic process of different conditions [25,26,27,28,29,30,31,32,33]. Numerous studies proposed indices for an objective gait assessment based on spatio-temporal and/or joint kinematics parameters [11,32,33,34,35,36], but only a few works used the information extracted from sEMG signals to this purpose [12,37,38,39]. In particular, in the work by Castagneri et al. [12], the asymmetry level of lower limb muscles in healthy, orthopedic, and neurological subjects was assessed by combining the SGA and CIMAP algorithm, suggesting that appropriate indices can be successfully used in clinics for an objective assessment of the muscle activation asymmetry during locomotion. In this context, the definition of a quantitative and reliable index for measuring the similarity of the dynamic muscle activation of a pathological subject with that of a reference population can be extremely useful for the assessment of the disease progression and for the evaluation of treatment outcomes. At this time, to the best of our knowledge, an index with these properties has not yet been presented in the literature.

The aim of this study is twofold. First, to present a Muscle Functional Index (MFI) that quantifies the similarity of the activation pattern of a muscle of a specific subject with that of the corresponding muscle of a reference population. Second, to present a Global Functional Index (GFI) to quantify the overall muscle activation similarity of a muscle set of a specific subject compared to that of a reference population. In this paper, we defined the two indices considering a reference population of healthy children and then we assessed the behavior of the proposed indices using two groups, one consisting of healthy children (not belonging to the reference population and referred to as “controls”) and a second one consisting of hemiplegic children.

## 2. Definition of the Indices

To assess the similarity between the activation pattern of the muscle(s) of a specific subject with respect to a reference population, we introduced two indices. The first index (MFI) quantifies the similarity of the activation intervals of a specific muscle of a subject with respect to the corresponding muscle of a reference healthy population. The second one (GFI) quantifies the overall similarity of the muscle activation patterns of a specific group of muscles with respect to those of the reference healthy population. The definition of the functional indices consists of two phases: (1) the characterization of the muscle activation of the reference healthy population and (2) the computation of the muscle functional indices. Figure 1 describes the various steps of each phase.

Both phases are based on the measure of similarity $Sim_{A,B}$ between the binary vectors A and B, of equal length (n bits), as calculated in Equation (1):(1)$$Sim_{A,B}=1-\frac{\sum_{i=1}^{n}\left|A_{i}-B_{i}\right|}{n}$$
where $A_{i}$ and $B_{i}$ are the values of the *i*-th bit in A and B, respectively. This measure evaluates the percentage of bits that are similar between vectors A and B and ranges from 0, if A and B are completely different, to 1, if the two vectors are equal.

### 2.1. Characterization of the Muscle Function Relative to the Reference Population

The characterization of the muscle function relative to the reference population consists of three steps: Section 2.1.1 extraction of the Principal Activations from the myoelectric signals collected on the subjects belonging to the reference population during the task to be studied, Section 2.1.2 description of the muscle activation modalities found in the reference population, and Section 2.1.3 calculation of the reference thresholds.

#### 2.1.1. Extraction of the Principal Activations from the Subjects Belonging to the Reference Population

First, the PAs of each muscle are extracted from all the subjects belonging to the reference population using the optimized version of the CIMAP algorithm [10]. To apply this algorithm, the sEMG signal acquired from a specific muscle is pre-processed as follows:The sEMG signal is segmented into separate gait cycles by using foot-switch signals and time-normalized to 1000 samples [13];The onset–offset activation intervals are detected by using a two-threshold statistical detector [14];The onset–offset activation intervals lasting less than 3% of the gait cycle are removed, while activation intervals separated by less than 3% of the gait cycle are joined together [40];Every *i*-th gait cycle is described through a vector containing m couples of onset–offset activation intervals ($ON_{i}$, $OFF_{i}$):(2)$$stride_{i}=\left\{ON_{i,1},OFF_{i,1},\ldots ,ON_{i,m},OFF_{i,m}\right\}$$
where m is the number of onset-offset activation intervals within the same gait cycle and generally differs from muscle to muscle.

The CIMAP algorithm [10] is then applied to all the gait cycles of each investigated muscle to obtain clusters showing similar muscle activation patterns. For each cluster, the strides belonging to right and left sides are separated and the prototype of each group is calculated as the median of the time instants ($ON_{i}$, $OFF_{i}$) (Figure 2a). The prototypes are coded as strings of 1000 bits (0 = no muscle activation; 1 = muscle activation). Then, the intersection of the corresponding cluster’s prototypes constitutes the PA (Figure 2b). At the end of this phase, every subject within the reference population is characterized by two PAs (one for each side).

#### 2.1.2. Description of the Muscle Activation Modalities Typical of the Reference Population

This step aggregates the information contained in the PAs extracted from the reference population for each investigated muscle. For a specific muscle, pairwise comparisons among a PA A and all the other PAs B in the reference population are performed using Equation (1). Then, the median of all the obtained similarity values ($R_{A}$) is calculated for the PA A as detailed in (3):(3)$$R_{A}=median\left(Dist_{A,B}\right),\forall B\neq A$$
where B represents every principal activation in the reference population except for A, and $Sim_{A,B}$ is the measure of the similarity as described by Equation (1).

After computing the *R* values for a given muscle ($R_{A})$, the maximum across the reference population $R_{max}$ is used to normalize every *R*-value:(4)$$R_{A,norm}=\frac{R_{A}}{R_{max}}$$

This normalization step allows for the obtaining of comparable values for different muscles, since the $R_{max}$ values generally differ in different muscles. At the end of this phase, a set of $R_{i,norm}$ values, representing the similarity of each *i*-th PA compared to the other PAs, describes the behavior of the population for each investigated muscle.

#### 2.1.3. Computation of the Reference Thresholds

A reference threshold $R_{Th}$ calculated for each muscle allows for the comparing of the muscle activation of a specific subject with that of the reference population. In particular, $R_{Th}$ was obtained as the 5th percentile of all $R_{A,norm}$ across the reference population. This means that 95% of the PAs in the reference population have a similarity higher than $R_{Th}$ compared to the other PAs. At the end of this phase, a reference threshold $R_{Th}$ is associated with each specific muscle.

### 2.2. Calculation of the Muscle Functional Indices

Given a subject that does not belong to the reference population, the extraction of the Muscle Functional Index (MFI) and the Global Functional Index (GFI) consists of three steps: Section 2.2.1 extraction of the PAs of the subject, Section 2.2.2 calculation of the MFI for every muscle, and Section 2.2.3 computation of the GFI.

#### 2.2.1. Extraction of the Principal Activations of a Subject

First, the two PAs (one for each side) are extracted for each muscle belonging to the muscle pool of interest using the optimized version of the CIMAP algorithm [10], following a procedure similar to that described above with respect to the reference population (Section 2.1.1 Extraction of the Principal Activations from the subjects belonging to the reference population).

#### 2.2.2. Calculation of the MFI for Every Muscle

For each muscle (left side and right side separately), the MFI is computed as detailed in Equation (5):(5)$$MFI=\frac{median\left(Sim_{S,A}\right)}{R_{max}},\forall A\;\mathrm{in}\;\mathrm{the}\;\mathrm{reference}\;\mathrm{population}$$
where $Sim_{S,A}\;$is the similarity among the PA of the subject S and all PAs in the reference population as obtained by Equation (1), and $R_{max}$ is the maximum R-value computed within the reference population.

The obtained MFI value can be compared with the corresponding reference threshold $R_{Th}\;$to assess the muscle function with respect to a reference population: an MFI value below the reference threshold represents an abnormal muscle function, while an MFI value above the threshold represents a muscle function comparable to that of the reference population.

#### 2.2.3. Calculation of the GFI

The GFI is the average of the MFI values (one for each muscle) for a given muscle pool of a specific subject (6):(6)$$GFI=\frac{\sum_{i=1}^{M}MFI_{i}}{M}$$
where M is the total number of observed muscles. The GFI quantifies the overall similarity of the activation patterns of a pool of muscles of a subject compared to the reference population.

## 3. Demonstration of the Applicability and Proper Behavior of the Indices

### 3.1. Subjects

In this study, we retrospectively analyzed gait data acquired from 105 school-age children [16,21]: 55 healthy children, without known neurological or orthopedic disorders, were used as reference population; 25 healthy and 25 hemiplegic children were used as test sets to evaluate the behavior of MFI and GFI. Table 1 reports the average anthropometric parameters of the populations.

### 3.2. Acquisition System and Experimental Protocol

To acquire sEMG, goniometric, and foot-switch signals, we used the wearable system STEP32 (Medical Technology, Turin, Italy), CE certified for clinical gait analysis. Participants were equipped bilaterally with:Three foot-switches (size: 10 mm × 10 mm × 0.5 mm; activation force: 3 N) attached beneath the heel, the first, and the fifth metatarsal heads of each foot;Two electrogoniometers (accuracy: 0.5°) attached to the lateral side of the knee joints;Five sEMG active probes in single differential configuration (two Ag-disks with a diameter equal to 4 mm per probe; inter-electrode distance: 12 mm; probe size: 27 mm × 19 mm × 7.5 mm) attached, after skin preparation, on the belly of each muscle. Specifically, we recorded signals from Tibialis Anterior (TA), Gastrocnemius Lateralis (LGS), Vastus Medialis (VM), Rectus Femoris (RF), and Lateral Hamstring (LH) muscles on both body sides. An expert user visually inspected signals to exclude the presence of crosstalk.

The signal amplifier had an adjustable gain (60–94 dB) and a 3 dB bandwidth ranging from 10 to 400 Hz. Gain was adjusted to fit the signal amplitude to the input dynamic range of the A/D converter as much as possible, but avoiding its saturation. The sampling rate was equal to 2 kHz, and signals were converted by a 12-bit A/D converter and stored on the hard disk of the host computer.

Figure 3 shows the acquisition system composed of the sEMG active probes, the foot-switch sensors, and the electrogoniometers. Figure 4 shows an example of sensor placement on a healthy subject.

Subjects walked barefoot at self-selected speed over a 10 m walkway, back and forth, for approximately 2.5 min. The experimental protocol conformed to the Helsinki declaration on medical research involving human subjects and was carried out in a clinical environment with continuous medical supervision. Subject assent and signed parental informed consent were obtained for each subject.

### 3.3. Signal Pre-Processing

Using the SGA routines included in the software of the acquisition system (which is CE certified), we obtained, for each lower limb, the following gait phases: Heel contact (H), Flat foot contact (F), Push off (P), and Swing (S) [13]. The sEMG signals were then segmented in separate gait cycles and time-normalized to the stride duration [13]. For healthy children, we considered only the gait cycles showing the typical sequence of gait phases (i.e., H, F, P, and S phase). For hemiplegic children, since a very small number of HFPS gait cycles was available, we analyzed the strides of the most represented sequence of gait phases of each subject [15,16].

A multivariate statistical filter was used to discard strides related to deceleration, acceleration, and changes of direction [13].

Subjects whose sEMG signals had an SNR value lower than 6 dB for even a single muscle were discarded from the analysis, since we considered the signal quality not suitable to warrant reliable data.

Finally, the onset–offset muscle activation intervals were detected for each muscle and each side through a double-threshold muscle activation detector specifically developed for gait analysis [14].

### 3.4. Characterization of the Muscle Function Relative to the Reference Population

The three steps described in the previous section were applied to data relative to the healthy children included in the reference population to obtain, for each muscle, the $R_{A,norm}$ value and the corresponding reference threshold $R_{Th}$.

### 3.5. Calculation of the Muscle and Global Muscle Functional Indices

Onset–offset activation intervals of the groups of healthy and hemiplegic children were used to compute the MFI for each muscle and subject, as well as the corresponding GFI. We believe that using a radar diagram is a simple and effective way for visually inspecting the behavior of a specific subject against the average behavior of the reference population. For each muscle, the MFI values can be compared with the corresponding reference threshold $R_{Th}$ of the healthy population, to visually check their similarity. Figure 5 shows an example of this representation used to inspect the behavior of specific subjects.

### 3.6. Statistical Analysis

We applied the Lilliefors test to assess the normality of the MFI and GFI distributions of hemiplegic children, both for the hemiplegic and the sound sides, and healthy children, both for the left and right sides. Based on the Lilliefors test result, a two-tailed paired Student *t*-test (α = 0.05) (in case of normal distributions) or a Wilcoxon signed-rank test (α = 0.05) (for non-normal distributions) was used to compare: (a) hemiplegic and sound side of hemiplegic children, (b) left and right side of healthy children. The statistical analysis was carried out using the Statistical and Machine Learning Toolbox of MATLAB^®^ release 2020b (The MathWorks Inc., Natick, MA, USA).

## 4. Results

The data of 31 children out of 105 were discarded due to the low SNR of the myoelectric signals: 15 children belonging to the reference population, 7 healthy children belonging to the control population, and 9 hemiplegic children.

An average of 168 ± 27 gait cycles were collected for each child of the reference population and an average of 167 ± 25 and 133 ± 35 gait cycles were collected for each child of the two test groups (healthy and hemiplegic children, respectively).

From the reference population of 40 healthy children, we obtained the following threshold values: $R_{Th}\;$= 0.86 for VM, $R_{Th}\;$= 0.83 for TA and RF, and $R_{Th}\;$= 0.78 for LGS and LH. Figure 5 reports the MFI and the GFI values for two representative subjects of the test set (panel a: a typically developing child; panel b: a hemiplegic child). The dotted red lines join the reference threshold $R_{Th}$ for each muscle. The blue lines join the MFI values of specific subjects. The radar diagram allows for the easy highlighting of muscles with an abnormal behavior and significantly simplifies the interpretation of the MFI and GFI values. As an example, Figure 5a shows that, for the representative healthy child chosen from the reference population, all the muscles (both sides) present an MFI value above the reference threshold $R_{Th},$ and the GFI values are equal to 0.98 and 0.95 for the left and right sides, respectively. Differently, for the hemiplegic child, whose results are reported by Figure 5b, only the Gastrocnemius Lateralis (LGS) and the Lateral Hamstring (LH) muscles of the sound side show MFI values above the reference threshold, meaning that their behavior is similar to that of 95% of the reference healthy population. For the hemiplegic side, the MFI values are below the respective thresholds for all the muscles studied. Tibialis Anterior (TA) and Gastrocnemius Lateralis (LGS) show MFI values close to those of the reference population, thus demonstrating minimal dysfunction of these muscles, while proximal muscles (RF, VM, and VL) show MFI values close to 0.6, thus demonstrating a noticeable dysfunction. Consequently, GFI values of the hemiplegic child are 0.73 and 0.66 for the sound and hemiplegic side, respectively. This demonstrates that, in this specific child, both lower limbs cannot be considered as normally functioning and, as expected, the hemiplegic side shows a more severe condition than the other side. In addition, the non-affected side may not be considered to have a normal function. Figure 5 shows how the two indices may quantify the degree of functionality of the investigated muscles in a specific subject, either normal or pathological. This is the most important use of the two indices in clinics. As an example, considering the healthy child (Figure 5a), the radar plot clearly shows that, on both body sides, all the observed muscles are above the threshold that represents the minimum value of the index obtained on the 95% of subjects belonging to the reference population. Hence, this specific subject may not be distinguished by 95% of subjects belonging to the reference population. Moreover, it is clear that, while on the right side all the five observed muscles show a value of the MFI index close to 1 (the best possible match to controls), on the left side the TA muscle shows a value of MFI that is only slightly higher than that corresponding to the threshold. This could be a suggestion for clinicians to investigate the behavior of the TA muscle more in depth, to decide whether to prescribe a rehabilitation program to the subject or simply to repeat the exam after 6–12 months, to document possible changes. More than one third of typically developing children show mild gait abnormalities when they undergo a gait analysis test; in most cases, these abnormalities have no clinical meaning or disappear when the subject grows up, but in some cases they are worthy of being treated, since they could cause problems in adulthood or in the elderly. Figure 5b is a clear example of how MFI can very simply indicate which muscles of the observed muscle pool show an altered activation. On the hemiplegic side, TA and LGS (dorsi and plantar flexors of the ankle) show an MFI value very close to the threshold, thus demonstrating their almost normal activation timing. On the contrary, RF, VM, and LH show MFI values definitely below the threshold, thus demonstrating that muscle timing is compromised at the level of proximal muscles, that control both knee (LH, VM, and RF) and hip (RF and LH). The left radar plot of Figure 5b shows that, on the sound side, the MFI value of TA is slightly lower than the threshold. More interestingly from a clinical point of view, it is evident that knee extensors (RF and VM) show MFI values close to those of the affected side, while the LH shows a timing compatible with that of 95% of the control population. The considerations above show how the MFI values can be very effective in outlining the inappropriate timing of some of the muscles belonging to the considered muscle pool. The value of the GFI quantifies how the timing of the considered muscle pool is close to that of the normal population.

Figure 6 shows the MFI value for each muscle and for each of the two test groups (Panel a: healthy children belonging to the control group; Panel b: hemiplegic children).

The dotted red lines join the reference thresholds $R_{Th}$ computed over the reference population for each muscle. The blue lines join the MFI values for each specific subject in the two test groups. It is evident that the MFI values for healthy children are mostly above the thresholds for all the muscles and both sides. Only in 2 out of 40 cases, on the left side, there are subjects whose MFI values relative to one or two muscles are slightly outside the behavior of 95% of the subjects belonging to the healthy population. For the hemiplegic group, on the contrary, the distribution of the MFI values is wider, showing an abnormal behavior, definitely more evident on the hemiplegic side. We included Figure 6 for two different reasons. First, for demonstrating that the control population (that was not used to obtain the threshold values) shows values of the MFIs that are almost always above the thresholds computed for each specific muscle, while this is not the case—as expected—for hemiplegic children. Hence, the behavior of MFI matches our expectations. Second, when considering hemiplegic children, it is evident that every subject shows a different pattern of MFI values, thus demonstrating the capability of the index to capture differences among different subjects.

Table 2 reports the mean, the first and third quartile of the MFI values for the two test groups and for each muscle and side. The last column of the table contains the values of the reference threshold $R_{Th}$ for the five muscles.

Figure 7 shows the boxplots of the MFI values for the five muscles observed in this study, for the test populations of healthy (in violet) and hemiplegic children (in orange), for the two sides, separately. Since most of the distributions resulted non-normal according to the Lilliefors test, the Wilcoxon signed-rank test was used for comparing the MFI values of each muscle separately. In particular, the values of the MFI are not statistically different between the left and right side of healthy subjects for all muscles (*p >* 0.05), as expected. For the hemiplegic children, MFI values are not statistically different between left and right lower limbs for all the muscles, except for the RF (*p* = 0.02). This is not surprising, because to compensate the deficiency of muscles on the anatomically affected side also muscles belonging to the non-affected side must modify and adapt their activation modality. Comparing the MFI values of each lower limb of the healthy children and the two sides of the hemiplegic population, it emerges that values are statistically different for all comparisons, except for: (i) the right side of the healthy children with respect to the sound side of the hemiplegic children for the RF muscle (*p* = 0.08) and (ii) the left side of the healthy children with respect to the sound side of the hemiplegic children for the LH muscle (*p* = 0.11).

Figure 8 reports the boxplots of the GFI values relative to the test populations of healthy and hemiplegic children (for the two sides, separately). The last row of Table 2 reports the mean, and the first and third quartile of the GFI values for the two test groups. Since all distributions resulted normal, the two-tailed Student *t*-test was applied for the comparison of the GFI values. In particular, the values of the GFI are not statistically different between the left and right side of healthy subjects (*p* = 0.47), as expected. Comparing the GFI values relative to each lower limb of the healthy population and the hemiplegic and sound sides of the hemiplegic children it is evident that values are statistically different (*p <* 0.001, all comparisons). Finally, when considering the GFI values of the hemiplegic and sound sides of hemiplegic children they are statistically different (*p* = 0.02), as expected.

## 5. Discussion

Since the 1980s, numerous studies demonstrated the utility of sEMG for investigating the muscle function in basic research as well as in clinics. Although numerous research studies relative to muscle physiology investigated the muscle function in healthy and pathological subjects at the level of basic research, the number of papers reporting of sEMG applications in clinics, which really improved the quality of patient management, is definitely more limited. Then, the body of knowledge available at this time, relative to research and clinical experiences carried out in the last 40 years, fully demonstrates the capabilities and outlines the limitations of sEMG.

A recent work [7] investigated the usage and barriers of sEMG in neurorehabilitation, by sending a 30-question survey to 52 experts on sEMG from different standpoints, countries, and backgrounds. Among the 18 questions for which a consensus higher than 75% was reached, some of them well relate to this work. Specifically, a consensus higher than 90% was reached on the following nine points:“sEMG provides information on the neuromuscular function that is not provided by other assessment techniques/tools in neurorehabilitation” (91%);“In clinical rehabilitation sEMG enhances the assessment and characterization of neuromuscular impairment in patients” (94%);“sEMG allows to evaluate the effects of non-invasive interventions designed to impact muscle activity” (91%);“sEMG may be useful to evaluate the appropriateness of the activation among muscles participating to a specific movement” (97%);“sEMG allows to outline the sequential timing of muscular actions during given movements” (100%);“sEMG allows to evaluate the appropriateness of the activation among muscles participating to a specific movement” (97%);“sEMG assessment can be performed as a stand-alone technique to complement/optimize gait/motion analysis” (100%);“Timing of muscle activations and their variability must be considered of utmost importance for clinical applications in neurorehabilitation among the EMG-derived variables” (100%);“The difficulty of performing sEMG data analysis and interpretation without specific education/training is a potential barrier to the employment of sEMG in clinical neurorehabilitation” (97%).

From the cited work, which is very recent and, at this time, unique in the field of sEMG applied in clinics, we can summarize three statements on which the consensus is total:sEMG is a necessary tool to obtain a deep insight into the role of different muscles during any kind of movement;sEMG can be used as a stand-alone technique or it should be used as a complementary tool in gait/motion analysis, principally considering the timing of muscle activation;Performing sEMG data analysis and interpretation, with the tools currently available, is a complex task that requires specific training.

Hence, we can infer that for encouraging the spread of the usage of sEMG in clinics, scientists working on basic sEMG research should develop tools as much as possible that are user-independent, widely tested, and useful in clinics for facilitating the interpretation of multiple sEMG recordings. This is the purpose of the methodology herein presented.

Since the 1990s, some of the authors of this paper devoted a large part of their research activities developing user-independent methods to facilitate the application of sEMG signal analysis in clinics. Briefly, we developed, among others, the following tools: (i) a double threshold statistical detector of muscle activation [14] (1998); (ii) a comprehensive methodology for user-independent gait analysis, in which sEMG plays a major role [41] (2012); (iii) an improved algorithm for the user-independent segmentation and classification of gait cycles from foot-switch signals [13] (2014); (iv) an algorithm for quantifying the gait impairment score based on fuzzy logic [11] (2017); (v) an algorithm for hierarchical clustering of muscle activations [9] (2017); (vi) a user-independent index for quantifying asymmetry in muscle activations [12] (2019). This paper is the natural extension of our previous work, since it describes the development of another two indices aimed at facilitating the usage of sEMG in clinics without requiring any user-dependent decisions. We stress that user-independent analysis is essential for assuring high repeatability of the obtained results among different laboratories.

We introduced two indices based on the Principal Activations, which numerically describe the muscle activations of a subject with respect to a reference population. We showed that the muscle activation of a subject may be quantitatively evaluated for a single muscle (muscle-specific index, MFI) as well as for a specific muscle pool (global index, GFI). Moreover, to easily identify those muscles that are not activated in a physiological way, we proposed the representation of the MFIs by means of radar diagrams. This kind of representation may be easily adapted to any number of observed muscles.

The proposed indices allow for quantifying the similarity of the muscle activation of a specific subject to normality, which is defined as the behavior of a reference population. The present work uses as the reference population a group of 40 typically developing children studied during gait by investigating a group of five lower limb muscles (TA, LGS, RF, VM, and LH). The choice of these muscles assures having at least a flexor and an extensor muscle for each of the three joints usually investigated in gait (ankle, knee, and hip), which we consider as a solution generally satisfactory from a clinical point of view [41].

Notice that, since each MFI value is computed considering the sEMG signal generated only by the muscle it refers to, it does not depend on the number of observed muscles. Once the MFI values are obtained for any specific number of muscles considered, the GFI value may be obtained as the average of all the computed MFIs.

MFI and GFI find their most important application in clinics, when used to compare the activation and coordination of the muscle pool of interest of a specific subject to that of a reference population, for identifying possible deviations from the “normality”, as Figure 5 shows. As an example, the left radar plot of Figure 5b shows that, on the sound side, the MFI value of TA is slightly lower than the threshold value. More interestingly from a clinical point of view, it is evident that knee extensors (RF and VM) show MFI values close to those of the affected side, while the LH shows a timing pattern compatible with that of 95% of the control population. A plausible clinical interpretation of this observation is that the compensatory effect on the sound lower limb, that is necessary to obtain an acceptable locomotion, principally involves knee extensors. This observation may play an important role in designing a rehabilitation protocol suitable to the needs of this specific subject, thus implementing a personalized-medicine-approach in rehabilitation. This specific capability of the proposed indices would help clinicians to develop a personalized rehabilitation program for each specific patient, thus improving the likelihood of success.

In this work, however, we compared rather extensively the behavior of the two indices when applied to a control population of typically developing children (different from that used to obtain the threshold values of the reference population) and a population of hemiplegic children. This was undertaken using two groups that are known to clearly differ in muscle activation patterns and coordination, only to give proof of the proper behavior of the indices. In fact, Table 2, Figure 6, Figure 7, and Figure 8 clearly show that MFI and GFI values differentiate the two sub-populations, demonstrating that the control population of typically developing children has a behavior that is always compatible with that of the reference population of typically developing children, while the group of hemiplegic children shows a clearly different behavior. These statements have been supported by proper statistics.

Notice that, even if 31 subjects out of 105 were discarded from the sample population, this does not limit the validity of the presented results. To the best of our knowledge, the data set used in this study is still the larger available database of sEMG and other gait-related signals describing the walking modalities of school-age children. Furthermore, several acquisitions were discarded solely due to poor adherence to the experimental protocol of the child, which is more likely to happen in younger subjects compared to older ones. Possible applications of objective indices describing the similarity of the activity of a single muscle or of a muscle pool belonging to a specific subject to the behavior of a reference normal population may have different applications in clinics. Indeed, indices can be used to demonstrate the anomalous function of a muscle during a specific task thus allowing the understanding of the causes of a motion abnormality, a necessary step for developing an effective rehabilitation program or to plan surgery. Moreover, indices quantifying the similarity of the muscle activation of a subject with reference to a matched normal population also allow for evaluating the effectiveness of a rehabilitation program over time, to document objectively the recovery of normal function at a single muscle level as well as at the level of a muscle pool.

Actually, when we refer to “normal population” in terms of gait, we must be aware that the concept of “normality” is often associated also to subjects that are affected by gait abnormalities that do not compromise noticeably their daily activities. It has already been reported that a fraction of typically developing children shows slight gait abnormalities that are not evident to the visual observation [21]. These abnormalities do not cause any specific limitation and hence are not reported to physical therapists or physicians for early correction. Nonetheless, it may not be ruled out that the possibility of such apparently negligible abnormalities, that may be already evident in childhood and in teenagers, could cause more severe problems to affected people in adulthood or in the elderly. These problems could range from an abnormal fall propensity to low back pain, and several other conditions. In this perspective, the availability of indices as MFI and GFI, as well as other indices quantifying the “quality” of gait [11,29,30,34,35], could allow for a relatively inexpensive and operator-independent screening of school-age children and teenagers, to identify slight gait abnormalities caused by poor muscle function or coordination and thus allowing for the definition of specific correction protocols.

Another important class of possible applications is represented by the follow-up of patients following a rehabilitation program after orthopedic surgery or for compensating gait abnormalities following acute, degenerative, and congenital conditions of the nervous system (i.e., stroke, multiple sclerosis, Parkinson’s disease, cerebral palsy, etc.). Even in longitudinal evaluations, the availability of user-independent and reliable indices is crucial to allow for an objective patient assessment [42].

Finally, another very interesting application of these indices relates to sports training. By considering a specific movement performed by top-level athletes, one could build a “top reference population” to be used for scoring the performance of less talented athletes and, possibly, suggesting training programs to improve their performance. At this time, we do not have experience in this application, but we do believe it is worthy of being investigated.

## 6. Limitations of the Study

Although we believe that the indices we propose may be beneficial in clinics, we are aware of some limitations.

First, to apply the proposed indices to patients, from childhood to the elderly, we need three reference populations: typically developing children, normal adults, and elderly people. This is a limitation, but overcoming it only requires collecting and processing data from subjects belonging to the populations of interest. We already started collecting data from normal adults and shortly we will extend the study to elderly people.

A second limitation of this study is the number of investigated muscles, which is in this paper restricted to only five lower limb muscles. For improving the possible impact of the indices in clinical gait studies, it could be desirable to consider more than five muscles for each subject side. As already stated, we are currently working towards obtaining a reference adult population and we are considering twelve lower limb and trunk muscles. We foresee considering a set of at least twelve muscles also in the elderly reference population. Recording twelve or more muscles from typically developing children was not considered in our previous study mainly for two reasons. (i) The time needed to prepare a subject would have been definitely longer, and it is challenging to keep children concentrated for long periods of time. (ii) Especially for younger subjects, it may be difficult to place a large number of EMG probes on the limbs, due to the limited size of limbs.

However, the limitations of this study we are aware of, which we can easily overcome by extending the number of reference populations and considering a larger muscle pool, are well counterbalanced by the principal strength of this approach. This is the possibility of easily developing reference data for every cyclic movement, such as biking, running, stairs climbing, upper limb reaching tasks, swimming, and many others. In fact, although we tested the two indices and their computation in gait, the algorithms may be easily adapted to every cyclic movement, thus considerably enlarging the range of possible applications. In fact, the CIMAP algorithm, that is crucial for the extraction of the PAs on which GFI and MFI are based, was optimized for cyclic movements in general [10].

## 7. Conclusions

This work describes two quantitative indices for evaluating muscle activation in gait studies or several other cyclic movements. The MFI is relative to the activation of a single muscle, part of an observed muscle pool, and GFI is relative to the entire muscle pool.

In this study, we described how to compute the two indices and we demonstrated their proper performance in gait studies, considering a reference population of 40 typically developing children in which we detected sEMG from five lower limb muscles. The extension of the application of these indices to subjects from childhood to old age only requires the definition of another two reference populations, namely, one of normal adults and a second one of physiological elderly subjects. We are currently working on obtaining these two reference populations. We increased the number of observed muscles in adults and in the elderly from five to twelve, to extend the applicability of the methodology to larger muscle pools.

In conclusion, MFI and GFI values can provide a quantitative and reliable evaluation of muscle activation for identifying the abnormal function of single muscles involved in different movements and in various populations.

Given the importance of the availability of data describing various reference populations, we believe that experienced researchers working in this field should share through public data repositories their data, to make it possible to other research groups working in rehabilitation and sports medicine to benefit from the open access to reliable data sets.

## Figures and Tables

**Figure 1 sensors-21-07186-f001:**
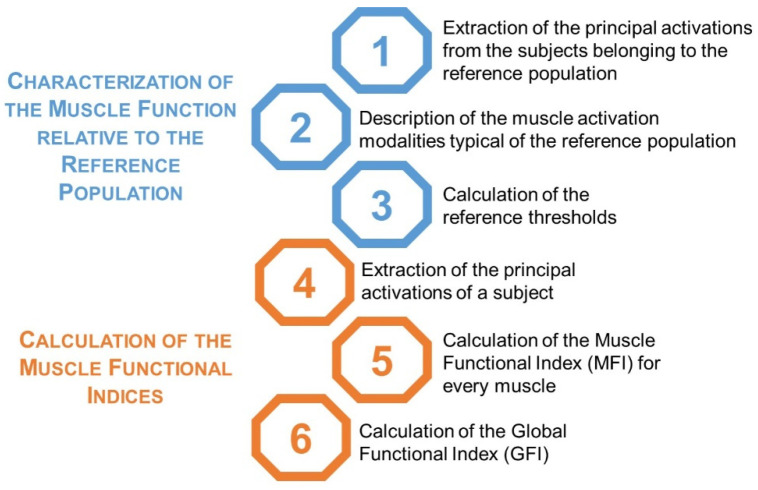
Pipeline for the definition of the muscle functional indices.

**Figure 2 sensors-21-07186-f002:**
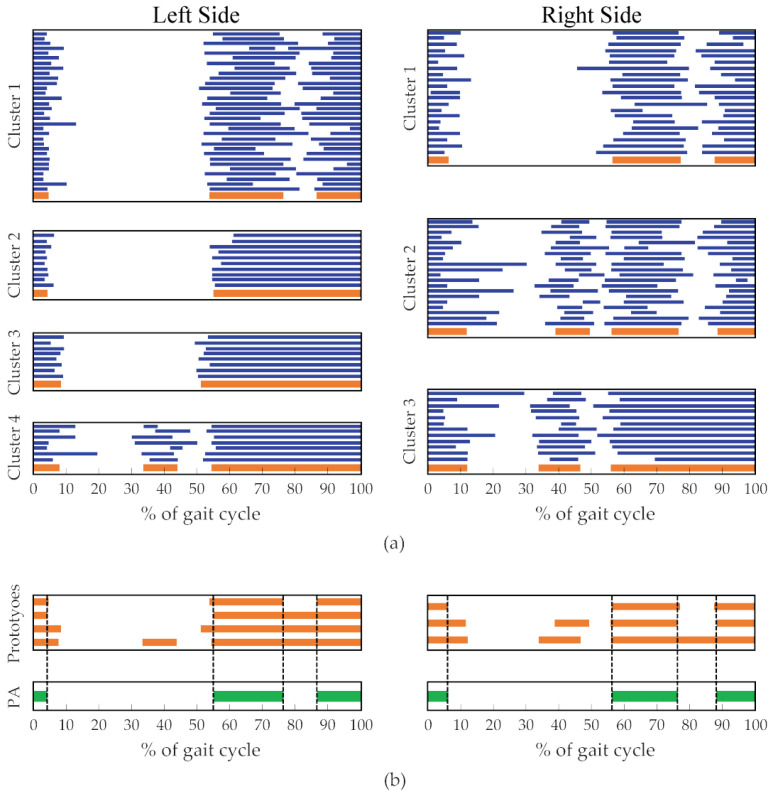
Example of PA extraction (Tibialis Anterior muscle). (**a**) Clusters resulting from the application of CIMAP. Strides belonging to the clusters are represented in blue, clusters’ prototypes are represented in orange. (**b**) PAs, obtained as the intersection of the cluster prototypes, are represented in green.

**Figure 3 sensors-21-07186-f003:**
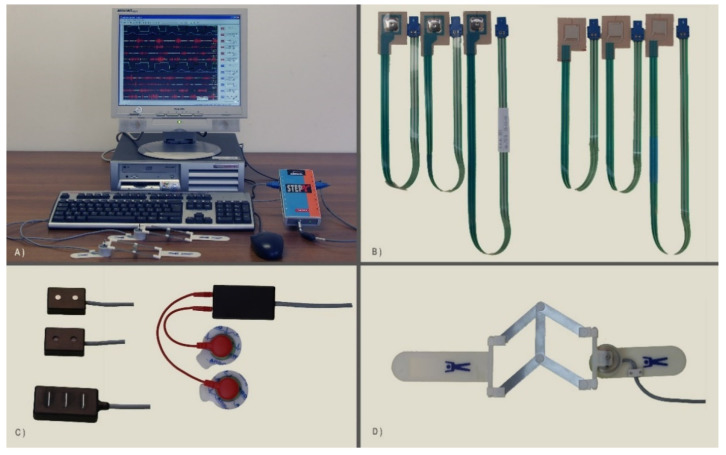
Details of the acquisition system: (**A**) the host computer, the patient unit, and two electrogoniometers; (**B**) two different kinds of foot-switches (on the left, a less sensitive set, for adults; on the right, a more sensitive set, for children); (**C**) different kinds of sEMG probes: two different versions of single differential probes (upper left); a three-bar double differential probe (lower left); a variable geometry probe (right); (**D**) a knee electrogoniometer.

**Figure 4 sensors-21-07186-f004:**
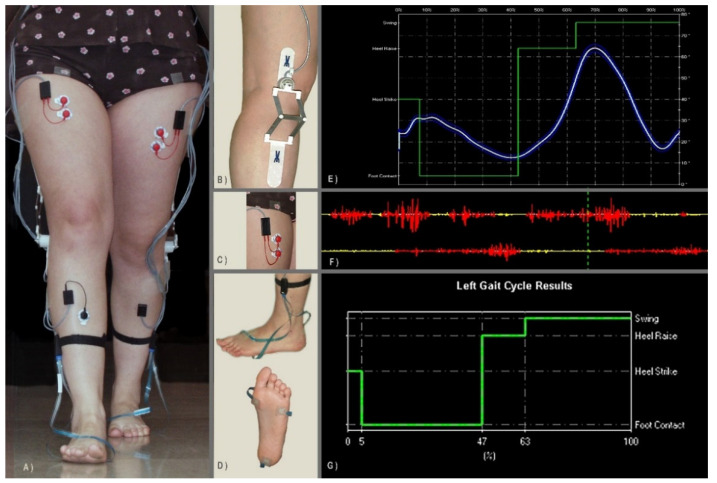
Sensor placement and recorded signals. SEMG active probes are positioned over the main muscles of the lower limb, bilaterally. Electrogoniometers are attached to the lateral aspect of the knee joints. Foot-switches are placed beneath the heel, the first, and the fifth metatarsal heads of each foot. (**A**) Subject performing an evaluation session. (**B**) Detail of the electrogoniometer attached to the lateral aspect of the knee to measure the knee joint angles during gait. (**C**) Detail of a variable geometry sEMG probe attached over the Rectus Femoris muscle of the subject. (**D**) Detail of the foot-switches attached underneath the first and fifth metatarsal heads and the heel (lower picture); how the foot-switches are attached to their connector (upper figure). (**E**) Example of the average variation of the knee joint angle over a given number of strides superimposed to the correspondent four-level coded foot-switch signal. (**F**) Example of two sEMG signals (Tibialis Anterior, upper trace; Gastrocnemius Lateralis, lower trace) collected during gait and processed by the user-independent activation detector: the yellow color means that the muscle is not electrically active and red color means that the muscle is electrically active. (**G**) Example of a four-level coded foot-switch signal: the four levels correspond to Heel strike (H phase), Flat foot contact (F phase), heel raise or Push off (P phase), and Swing (S phase); the sequence of foot-contact phases here represented corresponds to that observed in normal subjects during level walking.

**Figure 5 sensors-21-07186-f005:**
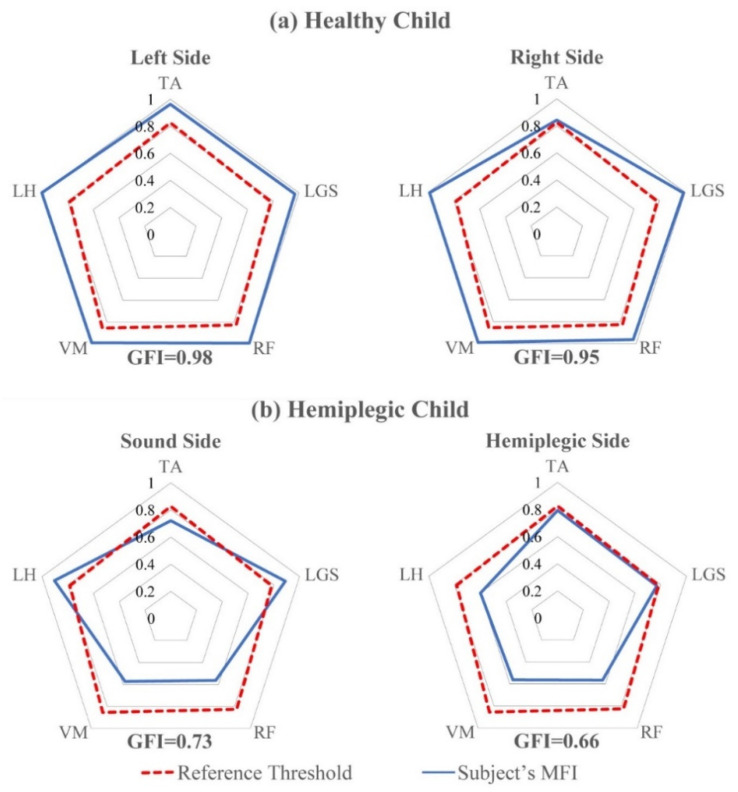
Radar diagram representation of MFI values for (**a**) a healthy child and (**b**) a hemiplegic child, both sides. The corresponding GFIs are reported under each diagram. The dotted red lines join the reference thresholds $R_{Th}$ for each muscle. The blue lines join the MFI values of the subject. Muscles: Tibialis Anterior (TA), Gastrocnemius Lateralis (LGS), Vastus Medialis (VM), Rectus Femoris (RF), and Lateral Hamstring (LH).

**Figure 6 sensors-21-07186-f006:**
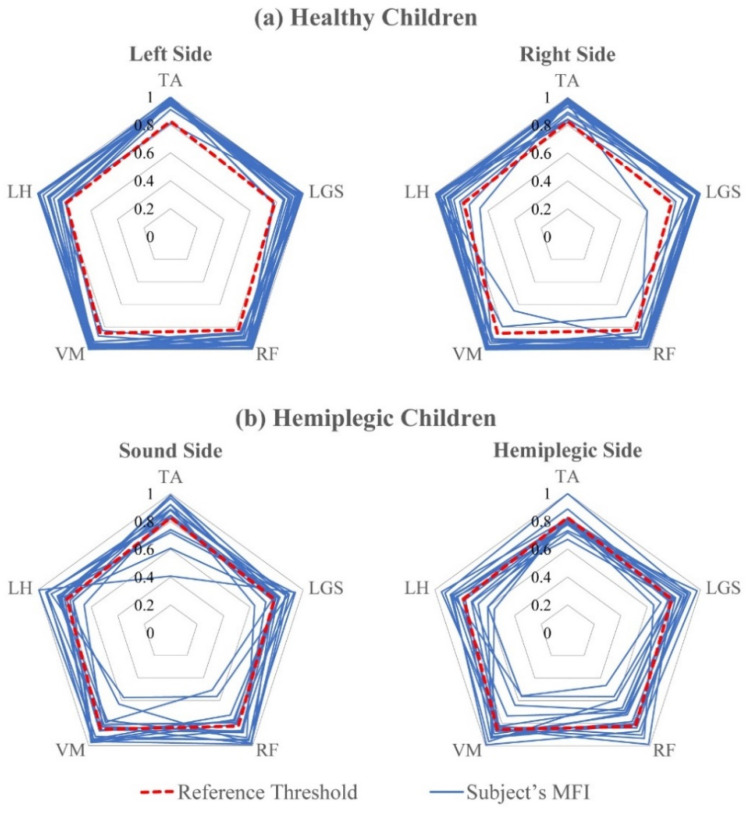
Radar diagram of MFI values for (**a**) the 18 healthy children and (**b**) the 16 hemiplegic children, both sides. The dotted red lines join the reference threshold $R_{Th}$ for each muscle. The blue lines join the MFI values for each subject in the two test groups. Muscles: Tibialis Anterior (TA), Gastrocnemius Lateralis (LGS), Vastus Medialis (VM), Rectus Femoris (RF), and Lateral Hamstring (LH).

**Figure 7 sensors-21-07186-f007:**
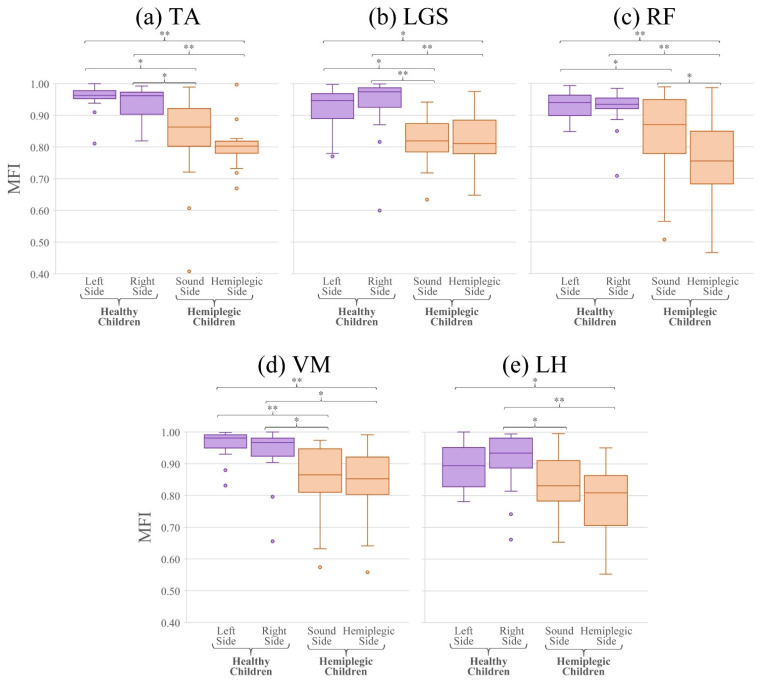
Boxplots of the MFI values of healthy and hemiplegic children of the test set, for the 5 muscles: (**a**) Tibialis Anterior (TA), (**b**) Gastrocnemius Lateralis (LGS), (**c**) Rectus Femoris (RF), (**d**) Vastus Medialis (VM), and (**e**) Lateral Hamstring (LH). Asterisks highlight statistically significant differences between groups or side (*: *p* < 0.05 and **: *p* < 0.001). Outliers are represented by circles.

**Figure 8 sensors-21-07186-f008:**
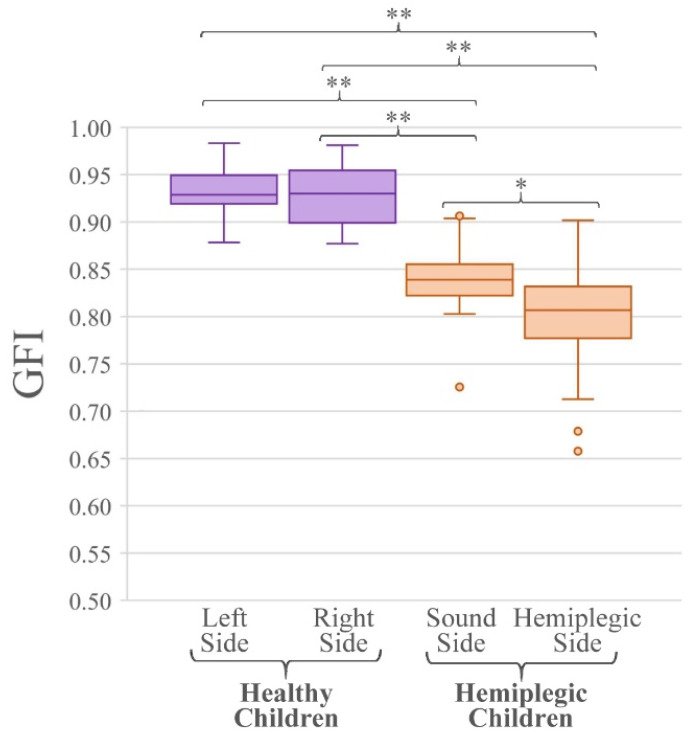
Boxplots of the GFI values of healthy and hemiplegic children of the test set. Asterisks highlight statistically significant differences between groups or sides (*: *p* < 0.05 and **: *p* < 0.001). Circles represent outliers.

**Table 1 sensors-21-07186-t001:** Anthropometric parameters of the populations.

	Number of Subjects	Age (Years) (Median and Range)	Gender ^1^	Height (cm) (mean ± S.D.)	Body Mass (kg) (mean ± S.D.)
Healthy Children (Ref. population)	55	9 (7–11)	28M/27F	133.1 ± 9.7	30.3 ± 6.2
Healthy Children (Test Set)	25	9 (6–11)	12M/13F	133.8 ± 9.1	31.1 ± 7.4
Hemiplegic Children (Test Set)	25	8 (4–14)	15M/10F	129.7 ± 18.8	30.2 ± 11.7

^1^ M = Male; F = Female.

**Table 2 sensors-21-07186-t002:** MFI and GFI values for the test groups (mean value [first and third quartile]) and the reference thresholds.

		Healthy Children		Hemiplegic Children		Reference Threshold $R_{Th}$
		Left Side	Right Side	Sound Side	Hemiplegic Side	
*MFI*	TA	0.96 [0.95 ÷ 0.98]	0.94 [0.90 ÷ 0.97]	0.83 [0.80 ÷ 0.92]	0.80 [0.78 ÷ 0.82]	0.83
	LGS	0.92 [0.89 ÷ 0.97]	0.93 [0.92 ÷ 0.99]	0.82 [0.78 ÷0.87]	0.82 [0.78 ÷ 0.88]	0.78
	RF	0.93 [0.90 ÷ 0.96]	0.92 [0.92 ÷0.95]	0.84 [0.78 ÷ 0.95]	0.75 [0.68 ÷ 0.85]	0.83
	VM	0.96 [0.95 ÷ 0.99]	0.93 [0.92 ÷ 0.98]	0.85 [0.81 ÷ 0.95]	0.82 [0.80 ÷0.92]	0.86
	LH	0.89 [0.83 ÷ 0.95]	0.91 [0.89 ÷ 0.98]	0.84 [0.78 ÷ 0.91]	0.78 [0.71 ÷ 0.86]	0.78
*GFI*		0.93 [0.92 ÷ 0.95]	0.93 [0.90 ÷ 0.95]	0.83 [0.82 ÷ 0.86]	0.80 [0.78 ÷ 0.83]	-

## Data Availability

Data presented in this study are available on request from the corresponding author.

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
