# Peer review of "Evaluation of Muscle Function by Means of a Muscle-Specific and a Global Index"

_sensors, 2021, doi:10.3390/s21217186_

Round 1

Reviewer 1 Report

General comments:

This is an interesting paper presenting an approach to quantify muscle activity in gait as measures by surface electromyography (S-EMG). Data of 105 children with and without neurologic disorders were retrospectively analysed. A “Muscle function index” (MFI) and a “Global Functional Index” (GFI) is introduced for characterizing deviations from normal muscle activation in gait. I do have several concerns with regard to motivation, methods, results, and discussion which should be addressed:

1)    The abstract starts with three arguments as to why clinical gait analysis would not be more commonly used. I agree only with the first one: feasibility and effort for preparing the subject. 2nd: User independent tools may be an issue but is not really clarified in this manuscript what a “user” really is. Is it the examiner or the person interpreting the data? This should be discussed in further details if the manuscript really wants to make a point here. 3rd: variability of muscle activation patterns. This per se cannot be an argument. Only that EMG still may not play the same role next to kinematics, kinetics, and clinical exam, since interpretation is difficult.
2)    In the introduction, only the large variation in EMG signals is given as motivation for implementing new numerical concepts for analysis. However, nothing is said about the physiologic background. The terms “muscle function”, “muscle activation”, and “EMG signal” seem to be interchangeable. In this regards the introduction needs to be specified.
3)    Methods: The mathematical recipes for EMG signal processing are described but not motivated. Terminology needs to be more precise. (see point above).
4)    The number of muscles may affect the index outcome significantly, and in this study only five muscles were used which could produce significant difference between CP and TD in context to typical clinical procedures where 8 muscles are measured.
5)    We may see differences between normal and abnormal (TD vs CP) but there may not be such differences between different pathological cases. Abnormal muscle index may stay in a fixed range. only the outcomes for TD versus Hemiplegia children are provided even though the authors addressed both problems in the limitations.
6)    Subjects/Data: 31 of 105 data sets had to be discarded (including 15 normals!) since the signal intensity was too low. A failure rate of 30% cases is an absolute “no-go” for clinical application. It seems that international quality standards have not been adhered to and may be very questionable for scientific analysis.
7)    Discussion: Authors repeatedly mention that these indices could be useful in clinical applications but yet there is no proper interpretation, which should be presented here.

Author Response

GENERAL COMMENTS:

This is an interesting paper presenting an approach to quantify muscle activity in gait as measures by surface electromyography (S-EMG). Data of 105 children with and without neurologic disorders were retrospectively analysed. A “Muscle function index” (MFI) and a “Global Functional Index” (GFI) is introduced for characterizing deviations from normal muscle activation in gait. I do have several concerns with regard to motivation, methods, results, and discussion which should be addressed:

The Authors thank Reviewer #1 for considering this paper as interesting. We will do our best to address each specific comment properly. We answered all Reviewer #1’s comments in the point-by-point response below, highlighting in a marked copy the changes made to the manuscript, following her/his suggestions.

SPECIFIC COMMENTS:

  1. The abstract starts with three arguments as to why clinical gait analysis would not be more commonly used. I agree only with the first one: feasibility and effort for preparing the subject.

On this first point, the opinion expressed by the Reviewer #1 confirms what we presented in the paper. Moreover, the fact that a potential barrier to the usage of EMG is represented by the fact that it is time-consuming is agreed by 77% of researchers and clinical practitioners interviewed in the very recent work cited as [7] in the manuscript.

 2nd: User independent tools may be an issue but is not really clarified in this manuscript what a “user” really is. Is it the examiner or the person interpreting the data? This should be discussed in further details if the manuscript really wants to make a point here.

Reviewer #1 is right, the definition of “user” was not explicitly introduced. We modified the introduction to clarify this point (page 1, lines 33-46; page 2, lines 47-77 of the revised manuscript).

3rd: variability of muscle activation patterns. This per se cannot be an argument. Only that EMG still may not play the same role next to kinematics, kinetics, and clinical exam, since interpretation is difficult.

 We completely agree with the reviewer that EMG still may not play the same role as kinematics, kinetics, and clinical exam since interpretation is less immediate. We think that, at least to some extents, this difficult interpretation can be related the high variability of muscle activation patterns. When considering the muscle activation patterns or the linear envelope obtained from a sEMG signal collected during gait, it was demonstrated that even a normal subject performing level walking at self-selected speed (the most natural walking condition) shows several different activation modalities of the same muscle (see Figure 2 of our manuscript and previously published works [8, 9, 10, 17, 19, 20, 23, 24], among others). This characteristic of muscle activation patterns makes it difficult the comparison among different subjects and the extraction of clinically relevant information. The subdivision of muscle activations during gait in primary (essential) and secondary activations (see Figure 2, extraction of principal activations, and for a complete discussion see Ref. [9]) greatly simplifies the understanding of causal relationships between principal activations of a muscle and their contribution to the movement. However, it must be emphasized that the contribution of secondary activations must not be disregarded. Our research group is currently investigating the causes and effects of secondary activations during gait, also at the level of muscle synergies. Results are promising, but not ready to be presented. Hence, in this study, we only consider principal activations, since this approach greatly simplifies the interpretation of muscle activation intervals in clinics and makes it easier to establish causal relationships between muscle activity and biomechanical outcomes.

  1. In the introduction, only the large variation in EMG signals is given as motivation for implementing new numerical concepts for analysis. However, nothing is said about the physiologic background. The terms “muscle function”, “muscle activation”, and “EMG signal” seem to be interchangeable. In this regards the introduction needs to be specified.

 We agree with Reviewer #1 that the terms “muscle function”, “muscle activation”, and “EMG signal” are different and that they must be used properly. In the introduction of the revised manuscript, we introduced the definitions of these terms (page 3, lines 100-115). Moreover, we carefully checked the revised version of the paper to make sure that these terms are used correctly.

  1. Methods: The mathematical recipes for EMG signal processing are described but not motivated. Terminology needs to be more precise. (see point above).

The rationale/motivation of the algorithms introduced to obtain MFI and GFI are now better described (page 3, lines 141-149). We carefully checked the revised paper for possible imprecise terminology and modified it, when necessary.

  1. The number of muscles may affect the index outcome significantly, and in this study only five muscles were used which could produce significant difference between CP and TD in context to typical clinical procedures where 8 muscles are measured.

Since MFI is obtained considering the sEMG signal generated only by the muscle it refers to, it does not depend on the number of observed muscles and hence it is possible to compute n different MFI values – one for each muscle – if n different muscles are studied. The database we used in this paper takes into account five muscles per limb, and hence we computed 10 different MFI values, one for each muscle and each body side. This is a limitation if one wants to use Reference Threshold values we reported in Table 2 (since only five muscles per side are considered), but the main purpose of this paper is to present a method that may be applied by different research groups on the number of muscles they are interested in, without limitations. Once MFI values are obtained, for any specific number of muscles considered GFI may be obtained simply as the average of MFI values. Again, the indices herein presented may be applied to as many muscles as the researcher is interested in.

We added in the Discussion of the revised manuscript a dedicated paragraph to emphasize this aspect (page 16, lines 517-519).

  1. We may see differences between normal and abnormal (TD vs CP) but there may not be such differences between different pathological cases. Abnormal muscle index may stay in a fixed range. only the outcomes for TD versus Hemiplegia children are provided even though the authors addressed both problems in the limitations.

 Thank you for raising this point. This allowed us for better explaining the concepts below (page 11, lines 357-380; page 12, lines 397-404).

  1. Subjects/Data: 31 of 105 data sets had to be discarded (including 15 normals!) since the signal intensity was too low. A failure rate of 30% cases is an absolute “no-go” for clinical application. It seems that international quality standards have not been adhered to and may be very questionable for scientific analysis.

We understand the concerns of Reviewer #1, but we believe that some considerations are necessary before stating: “It seems that international quality standards have not been adhered to and may be very questionable for scientific analysis”.

First, the signal-to-noise ratio of every signal depends, obviously, on the power of both signal and noise. In most biological signals, noise comes either from artifacts (i.e., motion artifacts) or from other biological sources that are active in the proximity of the detection probe (i.e., another active muscle, and in this case we refer this unwanted signal as crosstalk, or other possible signal sources). The noise generated by the electronics of the detection system is generally negligible. The signal intensity strictly depends on the thickness of the subcutaneous tissue and on the level of muscle activation. The thickness of the subcutaneous tissue may vary substantially from skinny people to people with a high body mass index. Hence, the signal-to-noise ratio in dynamic surface EMG is much more variable than ECG or EEG signals, since both signal amplitude and noise substantially change from subject to subject. More details about SNR measurement in dynamic sEMG may be found in our paper at http://dx.doi.org/10.1109%2FTBME.2011.2170687.

Second, the population consisted of 6 – 12 years-old kids, and not of adults. Kids do not adhere to instructions they receive relative to an attention-demanding task – as gait is – for more than 15 – 20 minutes, with a large variability due to their age and attitude. For ethical reasons, researchers decided not to spend more than 40 – 50 minutes to test every single kid, not to stress her/him too much. It must be kept in mind that this was an observational study and neither kids nor parents would have had any direct benefit from their participation. Moreover, even when it was obvious that a kid did not stick to the protocol (some of them were walking in weird ways, just for fun), researchers did not stop the test not to disappoint kids and parents.  Poor adherence to the protocol was more evident in younger kids, but also some older ones were often inattentive because other devices in the lab attracted their attention.

For the above reasons, the signal quality was inspected by two experienced researchers (MK, approximately 40 years spent studying sEMG signal, and VA, more than 15 years of experience in the field of sEMG detection and processing). The careful quality check performed on signals does warrant that the selected signals have a quality good enough for the purposes of the study. In this study, we report the number of kids discarded from the study due to low signal quality because we believe this is an important piece of information for other researchers planning to obtain a large database of sEMG signals from kids from 6 to 12 years old during self-selected speed gait.

 As a matter of fact, the signals selected for this study have already been recognized as of acceptable quality at an international level, since these data come from two databases previously described and published on highly respected international journals ([21] for normal subjects and [16] for hemiplegic children). We also underline that, to our best knowledge, at this time this is the larger available database of sEMG and other gait-related signals describing the walking modalities of school-age typically developing children.

 Nonetheless, we recognize that if Reviewer #1 made this comment, this means the considerations above must be added to the manuscript, even if in a synthetic form. Hence, we modified the paper accordingly (page 17, lines 545-551 of the revised manuscript).

  1. Discussion: Authors repeatedly mention that these indices could be useful in clinical applications but yet there is no proper interpretation, which should be presented here.

 The purpose of this paper was to present how to obtain two objective indices to describe the similitude of the principal activation of a specific muscle with that of a given reference population and to demonstrate that these indices can identify muscles that show an activation pattern that is different from that of a reference population. The MFI, at a muscle-specific level, and the GFI, at the level of the observed muscle pool. The mathematical construction of the two indices warrants that they represent the similarity of the activation patterns. To give evidence of the correct behavior of the indices, they have been applied to a control population and to a population of hemiplegic children. The MFI index of each considered muscle of the control population showed a value comparable with that obtained from the 95% of the reference population of healthy kids (the reference population and the control population of healthy kids were constituted by different subjects). Moreover, this study also shows that it is possible to identify, for each hemiplegic child, those muscles that show a behavior non-consistent with that of the reference population.

 To give an example of the clinical utility of the MFI index, as requested by Reviewer #1, in the discussion section of the revised manuscript we added the following sentence (page 11, lines 380-387).

Reviewer 2 Report

The purpose of the presented study were - in authors' opinion - the lack of user-independent tools in this area; the large variability of muscle activation patterns observed in healthy and pathological subjects. Numerical indices quantifying the muscle coordination of a subject could enable clinicians 14
to identify patterns that deviate from those of a reference population and to follow the progresses of the subject after surgery or completing a rehabilitation program. In this study, the authors presented two user-independent indices. First, a muscle-specific index (MFI) that quantifies the similarity of the activation pattern of a muscle of a specific subject with that of a reference population. Second, a global index (GFI) that provides a score of the overall function of a muscle set. These two indices were tested on two groups of healthy and pathological children with encouraging results. The authors maintain it will allow clinicians to assess the muscle function, identifying muscles showing an abnormal activation pattern, and associating a functional score to every single muscle as well as to the entire muscle set. These opportunities could contribute to facilitating the diffusion of surface EMG analysis in clinical apps.

I suggest some improvemements before acceptation:

-the paper should be checked by professional native speaker.

-the introduction section should be just like a critical/systematic review with presenting Evidence Based Status and literature review tables with level of evidence - PEDro and/or Cochrane or other methodology scales (see how to write the professional systematic review) 

-add much more photos from research projects with tool applications in paricipants, presentation of devices/equipment etc

-create the limitation chapter after discussion. The study has many weakness 

Author Response

GENERAL COMMENTS:

The purpose of the presented study were - in authors' opinion - the lack of user-independent tools in this area; the large variability of muscle activation patterns observed in healthy and pathological subjects. Numerical indices quantifying the muscle coordination of a subject could enable clinicians to identify patterns that deviate from those of a reference population and to follow the progresses of the subject after surgery or completing a rehabilitation program. In this study, the authors presented two user-independent indices. First, a muscle-specific index (MFI) that quantifies the similarity of the activation pattern of a muscle of a specific subject with that of a reference population. Second, a global index (GFI) that provides a score of the overall function of a muscle set. These two indices were tested on two groups of healthy and pathological children with encouraging results. The authors maintain it will allow clinicians to assess the muscle function, identifying muscles showing an abnormal activation pattern, and associating a functional score to every single muscle as well as to the entire muscle set. These opportunities could contribute to facilitating the diffusion of surface EMG analysis in clinical apps.

The Authors wish to thank sincerely this Reviewer for the time she/he spent revising this paper. We considered carefully all Reviewer #2 comments and, when necessary, the paper was modified as suggested.

SPECIFIC COMMENTS:

I suggest some improvements before acceptation:

  1. the paper should be checked by professional native speaker.

 A colleague that is English mother tongue (US) checked the paper, as requested by Reviewer #2, and she only made minor modifications.

  1. the introduction section should be just like a critical/systematic review with presenting Evidence Based Status and literature review tables with level of evidence - PEDro and/or Cochrane or other methodology scales (see how to write the professional systematic review)

The formats suggested by Reviewer #2 fit to review articles, in which a specific question is posed more often as a hypothesis to be tested. In this case, the literature is searched for all the papers which can help in finding arguments that suggest rejection or acceptance of the hypothesis, to be able, at the end, to make an informed and justified decision on whether the hypothesis is to be rejected or not. Hence, when dealing with this kind of reviews, we agree with Reviewer #2 that it is very important to classify each single cited article in terms of reliability of the adopted methodology, of the obtained results, and of the conclusions presented, to avoid being biased by articles that contain evident flaws. Cochrane or other similar scales are very useful to this purpose.

The manuscript we submitted for publication is not a review paper. The Introduction aims at presenting typical problems of instrumented gait analysis and sEMG processing during gait to readers that are supposed to have a strong technical background, but not necessarily a deep clinical or physiological knowledge. Hence, we presented and discussed the various relevant problems by citing works published mainly in the last two decades on well-respected international journals, to give the reader the opportunity to verify the statements we made.

We reviewed the Introduction and we made some changes to make it clearer, more oriented to the subject of the paper, and easier to be verified (page 1, lines 33-46; page 2, lines 47-77 of the revised manuscript).

  1. add much more photos from research projects with tool applications in paricipants, presentation of devices/equipment etc.

As suggested by Reviewer #2, some figures showing the instruments used in the study have been added in the revised version of the manuscript (Figure 3 and Figure 4).

  1. create the limitation chapter after discussion. The study has many weakness

A Limitation section has been added after discussion, as requested by Reviewer #2, by moving the discussion on the limitations of the study that were contained in the Discussion into a specific section (page 18, lines 589-616 of the revised manuscript).

Round 2

Reviewer 1 Report

My critique has been adequately addressed by the authors. 

Reviewer 2 Report

I accept the authors's responses